# Defective Induction of IL-27-Mediated Immunoregulation by Myeloid DCs in Multiple Sclerosis

**DOI:** 10.3390/ijms24098000

**Published:** 2023-04-28

**Authors:** Felipe von Glehn, Nathalie Pochet, Bibek Thapa, Radhika Raheja, Maria A. Mazzola, Sushrut Jangi, Vanessa Beynon, Junning Huang, Alessandro S. Farias, Anu Paul, Leonilda M. B. Santos, Roopali Gandhi, Gopal Murugaiyan, Howard L. Weiner, Clare M. Baecher-Allan

**Affiliations:** 1Neuroimmunology Unit-Department of Genetics, Microbiology and Immunology-Institute of Biology, University of Campinas, Campinas 13083-970, Brazil; felipe.vonglehn@unb.br (F.v.G.);; 2Ann Romney Center for Neurologic Diseases, Department of Neurology, Brigham and Women’s Hospital, Harvard Medical School, Boston, MA 02115, USA; 3Broad Institute of MIT and Harvard, Cambridge, MA 02142, USA; 4Partners Multiple Sclerosis Center, Department of Neurology, Brigham and Women’s Hospital, Harvard Medical School, Boston, MS 02115, USA

**Keywords:** relapsing-remitting multiple sclerosis (RRMS), IL-27, myeloid dendritic cells (mDCs), lymphoproliferative assay, Programmed Death Ligand 2 (PD-L2), Programmed Death 1 receptor (PD1), innate immune response

## Abstract

The purpose of this study was to examine whether myeloid dendritic cells (mDCs) from patients with multiple sclerosis (MS) and healthy controls (HCs) become similarly tolerogenic when exposed to IL-27 as this may represent a potential mechanism of autoimmune dysregulation. Our study focused on natural mDCs that were isolated from HCs and MS patient peripheral blood mononuclear cells (PBMCs). After a 24-h treatment with IL-27 ± lipopolysaccharide (LPS), the mDCs were either harvested to identify IL-27-regulated gene expression or co-cultured with naive T-cells to measure how the treated DC affected T-cell proliferation and cytokine secretion. mDCs isolated from HCs but not untreated MS patients became functionally tolerogenic after IL-27 treatment. Although IL-27 induced both HC and untreated MS mDCs to produce similar amounts of IL-10, the tolerogenic HC mDCs expressed PD-L2, IDO1, and SOCS1, while the non-tolerogenic untreated MS mDCs expressed IDO1 and IL-6R. Cytokine and RNA analyses identified two signature blocks: the first identified genes associated with mDC tolerizing responses to IL-27, while the second was associated with the presence of MS. In contrast to mDCs from untreated MS patients, mDCs from HCs and IFNb-treated MS patients became tolerogenic in response to IL-27. The genes differentially expressed in the different donor IL-27-treated mDCs may contain targets that regulate mDC tolerogenic responses.

## 1. Introduction

Multiple sclerosis (MS) is an autoimmune disease of the central nervous system (CNS) in which autoreactive pathological Th17/Th1 cells can cross the blood–brain barrier (BBB) and promote inflammation and damage to oligodendrocytes and neurons [1,2,3,4]. Dendritic cells (DCs) are also found in the cerebrospinal fluid (CSF), meninges, choroid plexus, and inflamed CNS parenchyma after BBB disruption [5]. DCs can damage surrounding tissue by releasing cytokines [6] and activating/polarizing T cells via combinations of costimulatory molecules and cytokines [7]. However, DCs can also induce T cell hypo-responsiveness or tolerance [8], as mature and usually proinflammatory DCs can become tolerogenic in response to local interaction with IL-10, TGFβ, or CCL18 [9,10,11]. Thus, mDCs can toggle between proinflammatory and anti-inflammatory activity by regulating the activation or differentiation of autoreactive effector or regulatory T cells [9].

Interleukin 27 (IL-27) is an IL-12 family cytokine that exhibits both pro- and anti- inflammatory activities depending on the target cell, as it can enhance CD8^+^ T-cell activity and also reduce the intensity of CD4^+^ T-cell responses and Th1/Th2/Th17 differentiation [12]. In mice, DC-derived IL-27 inhibits autocrine osteopontin (OPN) and reduces EAE severity [13]. Moreover, IL-27 directly induces mouse and human T cells to produce IL-10 [13,14], thus potentially inducing Tr1 regulatory cells. IL-27 signaling has been recently proposed to be required for DC induction of tolerance [15]. However, it remains unclear whether S-patient derived mDCs can respond to IL-27 with tolerogenic activity, and whether IFNb therapy, which can augment IL-27 [16], affects mDCs’ tolerizing capacity.

As the frequency of natural DCs is around 1% of peripheral blood mononuclear cells (PBMCs), there are not many publications that have studied activities exhibited by circulating human dendritic cells. Thus, most publications test the activities of monocyte-derived DCs that were induced to become Mo-DCs by culturing them with various cytokine mixtures [17]. In this report, we aimed to bypass the question as to whether the activities of induced Mo-DCs actually reflect those of ex vivo isolated DCs, especially since we wanted to test whether the DC function would change after 24 h incubation with the cytokine IL-27.

## 2. Results

### 2.1. mDCs Isolated from Untreated- and IFNb-Treated MS Patients Markedly Differ in Their Ability to Become Tolerogenic in Response to IL-27 Stimulation

To examine the capacity of healthy donor- and patient-derived mDCs to become tolerogenic and freshly isolated HC, untreated MS and IFNb-treated MS mDCs were isolated, pre-activated with media (IL-27, lipopolysaccharide (LPS), or LPS/IL-27 for 24 h), and then co-cultured with autologous naïve CD4 T-cells. After four days, proliferation was measured by incorporating ^3^H-Thymidine to examine the inhibitory capacity of the differentially treated different donor mDCs. As shown in Figure 1 and Appendix A, co-cultures established with mDCs from HCs (Figure 1A) and IFNb-treated MS patients (Figure 1B), after they had been pretreated with IL-27 or IL-27/LPS, exhibited significantly less T cell proliferation (approximately 30–40% less proliferation) than the co-cultures that received the mDCs pretreated with media or LPS alone. Strikingly, the T cell co-cultures that were established with the IL-27 pretreated mDCs isolated from untreated MS patients showed no reduction in T cell proliferation (Figure 1C). This suggested that the mDCs derived from untreated patients were defective in their tolerogenic response to IL-27. This is in direct contrast to the tolerogenic activity exerted by the IL-27-treated mDCs isolated from IFNb-treated patients.

As the IL-27 pretreatment induced the mDCs derived from HCs and IFNb-treated patients to inhibit the proliferation of co-cultured T cells, we next examined whether the cytokines secreted in the mDC:naïve T cell co-cultures would also show similar changes in cultures containing the tolerogenic mDCs (HCs and IFNb-treated patients). Using the Luminex multiplex assay, we examined how the addition of differentially preactivated mDCs affected the secretion of IFN-a2, IL-27, IL-22, IL-23, IL-17A, IL-17F, and IL-10. Interestingly, IL-27, IL-23, and IFN-a2, which are primarily secreted by DCs, were similarly secreted in co-cultures that contained the tolerogenic, IL-27-pretreated mDCs from HCs and IFNb-treated MS; however, they were strikingly different from their levels in the co-cultures containing the IL-27 pretreated mDCs from untreated MS patients. As shown in Figure 2, the co-cultures containing the tolerogenic, IL-27-pretreated mDCs showed similar patterns of secretion of IFN-a2 and IL-27, while the co-cultures’ IL-27-pretreated mDCs from untreated patients did not show similar secretion patterns. In contrast to the mDC-derived cytokines, the pattern of secretion of T cell derived cytokines, IL-17A, IL-17F, IL-22, and IL-10, was most similar for the co-cultures containing mDCs from the untreated- and the IFNb-treated MS patients. Thus, the similarlity of the T cell-derived cytokines reflect the disease state, distinguishing between HC and MS patients (untreated and IFNb treated), while the innate cytokines reflect the tolerogenic capacity of the IL-27-pretreated mDCs (HCs and IFNb-treated patients). Additional effects of IL-27-induced tolerogenic mDCs were indicated by a reduction in T cell secretion of TNF-alpha, sCD137, and sFAS ligand, with strong increases in the secretion of IL-10 (Figure 2; Appendix A).

### 2.2. IL-27 Induces PDL2 and IDO1 in Human Ex Vivo mDCs

To begin to elucidate the mechanism by which IL-27 priming induces mDCs to become tolerogenic, we tested whether HC-derived mDCs respond to IL-27 through the induction of inhibitory immune molecules, e.g., PDL1, PDL2, and ICOS-Ligand. It is known that LPS-activation not only matures poorly stimulatory immature DCs but also stimulates human monocyte-derived DCs to increase their expression of IDO1 and B7 family molecules [17,18,19]. Furthermore, as IL-27 is a potent inducer of IL-10 [20], we examined if IL-27-mediated effects were mirrored by IL-10.

PD-L2 is expressed at very low levels on ex vivo mDCs (1.08 ± 0.68%, media) (Figure 3A,B top). Yet, mDC expression of PD-L2 was increased after a 24-h stimulation with IL-27 (2.59 ± 2.20%, *p* = 0.02) or IL-10 (5.76 ± 2.18%, *p*= 0.007) (Figure 3B top). Although stimulation with maturation-inducing LPS resulted in much a higher expression of PD-L2 (21.73 ± 15.81%), the co-administration of IL-27 or IL-10 with LPS even further enhanced PD-L2 expression over that seen by LPS treatment alone (26.23 ± 16.74%, *p* = 0.01 and 37.65 ± 13.73%, *p* = 0.03 respectively). Yet, the mDCs did not secrete more IL-10 when stimulated with IL-27 than when stimulated with media alone (12.87 ± 22.19 pg/mL vs. 5.88 ± 10.75 pg/mL, respectively). Similarly, the IL-10 that was strongly secreted in response to LPS stimulation (55.02 ± 35.87 pg/mL, *p* < 0.001) was not augmented by the co-administration of IL-27 (58.26 ± 47.36 pg/mL) (Figure 3C).

IDO is expressed at much higher levels than PD-L2 on ex vivo mDCs (17.06 ± 5.76%, Figure 3B). The 24-h treatment with IL-27 strongly enhanced mDCs’ expression of IDO1 (37.06 ± 15.7%, *p* = 0.004), while IL-10 treatment inhibited IDO1 expression (9.38 ± 4.9%, *p* = 0.02). Again, although LPS alone strongly induced mDC expression of IDO, the co-administration of IL-27 further enhanced IDO expression (89.27 ± 12.42% LPS vs. 93.46 ± 5.09% LPS/IL-27; *p* = 0.02) while the co-administration of IL-10 reduced the mDC expression of IDO (61.73 ± 16.9% LPS/IL-10, *p* < 0.001).

These studies indicated that IL-10 and IL-27 do not have the same effects on mDC expression as on regulatory molecules. More specifically, IL-10 or LPS/IL-10 induced the highest expression of PD-L2 on mDCs, while IL-27 or LPS/IL-27 induced the highest expression of IDO on mDCs. However, IL-27 stimulation did increase mDC expression of both PD-L2 and IDO1, while IL-10 stimulation only increased PD-L2 expression. Although IL-27 often induces the secretion of IL-10, the finding that IL-27 and IL-10 exert opposing effects on mDC expression of IDO suggests that IDO1 expression may be regulated by IL-27 independent of IL-10.

In examining whether IL-27 mediated these effects via IL-10, mDC cultures were supplemented with Ig or anti-IL-10. Neutralizing IL-10 inhibited the ability of IL-27 to induce PD-L2 (12.96 ± 7.51% LPS/IL-27/Ig vs. 8.51 ± 4.73% LPS/IL-27/anti-IL-10, *p* = 0.009) (Figure 3D) but had no effect on the IL-27-mediated induction of IDO.

### 2.3. mDCs Isolated from Untreated Patients with MS Show an Altered Gene Expression Response to IL-27

To identify if HC-derived and untreated (UnRx’d) patient-derived mDCs differ in intracellular signaling pathways that could drive tolerogenesis, we examined whether the different donor mDCs showed similar transcriptional responses to IL-27 treatment. However, as differential expression of IL-27RA could underlie potential differences in the capacity of patient-derived mDCs to respond to IL-27, we first determined that the patient- (untreated) and HC-derived mDCs expressed similar levels of IL-27RA (Figure 4A, left). In addition, we also found that mDCs from HCs and untreated MS patients expressed similar levels of IL-27RA in intracellular stores and thus have a similar capacity to respond to IL-27 (Appendix A). Similarly, as the differential induction of IL-10 could modify IL-27 responses using the different donor mDCs, we show that LPS and IL-27/LPS induced similar levels of IL-10 mRNA in the untreated MS patient- and HC-derived mDCs. Importantly, these data suggest that, since the capacity of the LPS- or IL-27/LPS-stimulated mDCs to express IL-10 did not differ between the mDCs isolated from HCs or untreated patients with MS, it is not a basis for the differential induction of tolerogenic activity by IL-27.

As both HC- and patient-derived mDCs similarly expressed IL-27RA and IL-10, we next tested whether the different donor mDC showed a differential induction of specific pro- and anti-inflammatory genes in response to IL-27. Using qRT-PCR with a panel of specific primers, we determined whether IL-27 treatment altered the expression of known pro- and anti- inflammatory genes in the context of LPS maturation signals. In addition to IDO1 and PD-L2, which we showed were regulated at the protein level by IL-27, we also tested whether IL-27 affected the level of the LPS-induced transcription of PD-L1, SOCS1, IL6R, CFS2BR (GM-CSF receptor), STAT5A, STAT1, and STAT3 within the different source mDCs. As shown in Figure 4B, stimulating the mDCs with IL-27/LPS compared to LPS stimulation alone resulted in a significant induction of mRNA for PD-L2 (*p* = 0.02), IDO1 (*p* = 0.02), and SOCS1 (*p* = 0.008) in HC mDCs, while the same stimulation of untreated-patient-derived mDCs resulted in a significantly higher expression of IDO1 (*p* = 0.007) and IL-6R (*p* = 0.04). In comparing the responses of HC- vs. untreated MS-derived mDCs to treatment with IL-27/LPS, the patient mDCs expressed significantly more STAT1 (LPS+IL-27 *p* = 0.04) and STAT3 (LPS+IL-27 *p* = 0.02) than the similarly treated HC mDCs. These data indicate that the untreated patient mDC responds to IL-27 (with LPS) with an increased expression of proinflammatory molecules, which may alter the ability of the patient-derived mDCs to transition into a tolerogenic state.

### 2.4. Circulating mDCs Exibit Different Gene Expression Signature Regarding Disease and Treatment Status

Previous studies have shown that patients with MS that are treated with IFNb have increased serum levels of IL-27 [16] and have increased SOCS1/decreased STAT1 expression in their immune cells [21]. To determine if we can identify gene expression signatures in mDCs that allow them to respond to IL-27 with the induction of tolerogenic activity, we used Nanostring RNA hybridization, which quantitates the actual number of transcripts in each sample. To identify genes involved in regulating mDC response to IL-27, we obtained RNA samples from mDCs that were sorted using fluorescence-activated single-cell sorting (FACS) from HC (n = 4), UnRx’d MS patients (n = 4) and IFNb-treated MS patients (n = 4). A third of the cells were immediately solubilized to examine their ex vivo gene expression (time 0), while the remaining cells were stimulated with IL-27 or media alone for 24 h before being harvested for RNA to allow us to examine change in gene expression from ex vivo values (see Figure 5A,D).

As only the HC- and IFNb-MS-derived mDCs become tolerogenic with IL-27 treatment, the next question was to identify the genes that are similarly induced by IL-27 in HC- and IFNb-MS-derived mDCs but which are distinct from those induced in the mDCs from untreated MS patients that do not become tolerogenic. The RNA from the different quadruplicate biological replicate samples were run on a defined nanostring codeset (Human Immunology v2-Nanostring Technologies). The analysis of the transcripts that changed with IL-27 treatment in the different donor mDCs groups is presented in the heatmap analysis in Figure 5A. The analysis resulted in six modules of genes that show unique regulation with the donor type and with IL-27 treatment. The modules 1 and 6 highlight the responses to IL-27 that differ between HC and MS patients regardless of treatment; these are likely more related to fundamental disease processes than the transition to a tolerogenic function. Similarly, modules 2 and 5 represent genes that are expressed at fundamentally different levels in mDCs from IFNb-treated patients compared to the mDCs from both HCs and untreated patients. These genes likely represent IFNb response genes. Lastly, modules 3 and 4 represent the genes that were similarly affected by IL-27 in the mDCs from HCs and IFNb-treated patients. Thus, the list of genes in modules 3 and 4 may contain the genes and pathways potentially involved in regulating the capacity of an mDC to become tolerogenic. In addition to the various groupings/modules of the gene expression patterns, it is possibly not surprising that the subjects are clustered by disease and treatment status in PCA analysis (HC vs. IFNb-MS vs. Untreated MS), again indicating that they express different gene signatures (Figure 5B).

To focus on the individual genes relevant to our study, the changes in mRNA expression of *STAT1*, *STAT3*, *IDO1*, and *PD-L2*, as detected by the nanostring data, were graphed individually (Figure 5D and Appendix A). mDCs freshly isolated from IFNb-treated patients showed significant greater expression of the transcription factor *STAT1* mRNA (*p* < 0.01) and a trend to increased *STAT3* as compared to mDCs from HCs and UnRx’d MS patients. This suggests that the cells may have been activated in vivo through elevated IL-27 serum levels that are induced by IFNb, as *STAT1* and *STAT3* are the main transcription factors activated by IL-27 [16]. Strikingly, after cultivating mDCs in media alone for 24 h, *STAT1* expression from IFNb-treated patients decreased to basal levels; however, they further increased when cultured with IL27 (z-score −8.58, *p* = 0.002 and z-score 1.76, *p* = 0.15, respectively). Confirming our protein and qPCR findings, the 24 h stimulation with IL-27 also strongly increased the expression of both PD-L2 (*PDCD1LG2*) and IDO1 in mDCs from HCs (z-score 13, *p* = 0.006 and z-score 6.6, *p* = 0.009, respectively), while these genes only showed a non-significant trend to increase in the mDCs of patients with MS (Appendix A).

## 3. Discussion

We demonstrated that, in response to IL-27, healthy donor mDCs (i) increase both their PD-L2 and IDO1 expression and (ii) exhibit a tolerogenic phenotype when co-cultured with CD4^+^ T-naïve cells. The IL-27-mediated induction of PD-L2 expression on healthy donor mDCs was shown to involve IL-10. In contrast to the mDCs from healthy donors, mDCs isolated from untreated patients with MS responded to IL-27 with the induction of IDO1 but did not increase their expression of PD-L2. Importantly, the mDCs from untreated patients also did not become tolerogenic in response to IL-27 treatment. Since previous studies indicated that patients with MS that are treated with IFNb exhibit marked increases in serum IL-27 [16], we determined whether the mDCs from the IFNb-treated patients differed from those isolated from patients that were not receiving any treatment. Strikingly, in contrast to the mDCs isolated from the circulation of untreated patients, the mDCs isolated from the IFNb-treated patients did become functionally tolerogenic after treatment with IL-27. Excitingly, these data support the hypothesis that IFNb treatment enables patient mDCs to develop the capacity to respond to IL-27 with tolerogenic function.

These data are also the first to demonstrate that IL-27 or IL-10 stimulation increases the expression of PD-L2 on human healthy donor mDCs. PD-L2, a B7 molecule family member, is known to inhibit T cell responses as it inhibits T cell proliferation by blocking cell cycle progression (G0/G1 phase) and cytokine production after activating PD-1 on the T-cell surface [22,23]. PD1 knockout mice develop a variety of autoimmune pathologies, highlighting the role of PD-1 as a negative regulator of immune response [24]. The induction of PD-L2 on the mDC may contribute to the induction of tolerogenic function. PD-1/PD-L1 interaction has recently been shown to offer protection from severe EAE in which IL-27-primed naïve T cells underwent reduced Th17 differentiation [25]. It has been shown that endothelial cells in the human brain normally express low levels of PD-L2 constitutively and respond to inflammation, with increased expression of PD-L2, which is hypothesized to inhibit the activation of human T cells as they migrate into the CNS [26]. Strikingly, the endothelial cells in the brain lesions in patients with MS have been found to express reduced levels of PD-L2, suggesting that the disease may be associated with a defect in PD-L2-mediated regulation [26].

IDO1 can also function to block T cell proliferation and does so by depleting tryptophan from intracellular and extracellular sources or by generating tryptophan degradation metabolites (mainly 3-hydroxy kynurenine, quinolinic acid, and kynurenic acid) that can also be neurotoxic [17,27]. A recent study demonstrated that IDO1-positive, monocyte-derived DCs expressed increased levels of B7 family members, including PD-L2, compared to IDO-negative-induced mDCs. The intracellular mechanisms linking increased IDO and PD-ligand expression were proposed to potentially mediate the downstream effects of IDO1 in suppressing T cells [17]. Importantly, in our report, we examined how IL-27 affected purified circulating myeloid dendritic cells in contrast to the majority of reports utilizing monocyte-derived/generated dendritic cells. Natural circulating mDCs are primarily derived from fms-related tyrosine kinase 3 ligand (Flt3L) stimulation [28] that induces the normal development of mDCs and plasmacytoid DCs while GM-CSF-differentiated DCs have a more immunogenic and inflammatory phenotype [29]. Thus, these MoDCs or iDCs and natural circulating DCs may differ significantly due to STAT5 signaling in MoDCs [30], while Ftl3L utilizes STAT3 signaling [31] that promotes the transcription of IDO1 [32] and inhibits NF-KB transcriptional factors [33,34]. Interestingly, we demonstrated that IL-10 simultaneously increased PD-L2 and decreased IDO1 expression in mDCs, and this effect was abrogated after blocking IL-10 from the culture (Figure 4). We speculate that it could be part of a regulatory feedback mechanism between mDCs and T-cells to control inflammation. However, as the mDCs isolated from untreated MS patients and healthy donors exhibited a similar induction of IDO1 in response to IL-27 but differed significantly in their ability to inhibit T cell proliferation, it is likely that the production of IDO1 alone is not responsible for the ability of mDCs to exert tolerogenic activity.

Analyzing the cytokines secreted in the tolerogenic and non-tolerogenic mDC co-cultures suggests that, unlike the pattern of cytokines that are derived from the T cells (high IL-10, IL-22, IL-17A, and IL-17F), it is the pattern of innate/DC-derived cytokines that segregates the samples along their tolerogenic outcome. The supernatants from the mDC-naive T co-cultures of cells from RR-MS patients, regardless of treatment status, did not demonstrate the upregulation of IL10 and downregulation of the proinflammatory cytokines IL-17F, TNF-α, and IL-23, as well as the lymphocyte activation marker 4-1BB (CD137), observed in the healthy donor mDC co-cultures. In fact, the assay of untreated- or IFNb-treated patient cells are highly similar in their secretion of T-cell-derived cytokines and strikingly different from the pattern of T cell cytokines produced in healthy donor cultures. Thus, it may be that the T cell cytokine pattern may generally distinguish patient-derived samples from healthy donor-derived samples.

In contrast to the T cell cytokines, the pattern of the innate/DC-produced cytokines (IL-27 and IFN-a2) actually align the tolerogenic healthy donor samples with the tolerogenic IFNb-treated patient samples. These data suggest that the enhanced secretion of IL-27 and/or IFN-a2 may be pivotal for, or indicative of, the tolerogenic activity of mature human mDCs. As only the IL-27-pretreated mDCs that exhibit tolerogenic activity also exhibited the enhanced secretion of IL-27 in subsequent co-culture (from healthy donors and IFNb-treated patients, Figure 2), it suggests that there may be an IL-27 feed-forward loop that does not occur in the mDCs isolated from untreated patients. In addition, as the tolerogenic co-cultures also contained signficant levels of IFN-a2, which is produced by innate cells and is known to regulate T-cell proliferation and apoptosis [35], the ability of IL-27-modulated mDCs to control T cell expansion may also involve IFN-a2 secretion. However, as the mDCs isolated from IFNb-treated patients are not tolerogenic unless they are pretreated with IL-27, the in vivo IFNb treatment may only confer the patient-derived human mDCs with the capacity to respond to IL-27, after a higher expression of IFN-a2 and IL-27. PCA analysis of the hybridization gene expression assay of 594 target genes indicated that naturally circulating mDCs from HC, UnRx’d, and IFNb-treated MS patients express different gene signatures (Figure 5B). Upregulation of the transcription factors *STAT1* and *STAT3* in natural circulating mDCs from patients treated with IFNb when compared to HC and UnRx’d MS (Figure 5C) indicates a tolerogenic effect of serum IL-27. A decrease in *STAT1* and *STAT3* mRNA expression after in vitro culture with media in IFNb-treated MS and not in HC and untreated MS was associated to their increased expression in all groups after IL-27 in vitro exposure, which confirms our results and is in accordance with the literature (Figure 5D) [21].

Changes in viability of isolated and cultured cDCs have been underappreciated in the published literature and could be a confounding variable in long-term in vitro studies. In their capacity to induce T cell hypo-responsiveness, tolerogenic DCs have been proposed either to be unable to provide co-stimulatory signals or to actively provide dominant negative signals [36]. Fundamentally, there is a concern whether the mDCs are equally viable and functional after the 24 h of pretreatment, or whether the IL-27-treated DCs are non-viable and thus are non-stimulatory rather than tolerogenic. As even tolerizing DCs are known to initially allow a certain amount of T cell proliferation [37], one way to measure DC viability would be to ascertain the capacity of each DC population to induce the initial activation of naïve T cells, a process requiring immunological synapse formation and TCR crosslinking [38,39]. Thus, we theorized that, if the cDC1/cDC2 underwent significant apoptosis in our 24 h pretreatment cultures, they would exhibit a severely reduced capacity to mediate initial T cell activation, which is associated with their release of sCD137, sFASL, and extracellular Granzyme B. The data in Appendix A show that the differently treated HC, MS-IFNb, and MS-UnRx’d mDCs exhibit an equal capacity to induce co-stimulation and synapse formation, as indicated by the similar and high detection of T cell-derived sCD137, sFASL, and GzmB. Thus, we believe that these data indicate that all of the 24 h pretreated DCs are able to similarly engage and activate the T cells to a certain extent. Whether they exhibit altered viability by the end of the four-day culture is an important question that we will include in our future studies. Regardless, if the UnRx’d MS mDCs lack tolerogenic activity because they preferentially undergo apoptosis in response to IL-27, then this in itself would be a striking mechanism.

In conclusion, the data provided in this study suggest that IFNb may regulate IL-27 responsiveness, and that the IL-27-mediated induction of PD-L2, IL-27, and IFN-a2 may play a role in the induction of tolerogenic DC function. Overall, these data also indicate that, in MS, the mDCs may be dysfunctional in their capacity to respond to IL-27 with tolerizing function and thus may contribute to MS pathogenesis.

## 4. Methods

### 4.1. Patients

In the first phase, we analyzed mDCs isolated from 18 healthy controls to study their capacity to regulate the in vitro activation and polarization of T cells. We performed a cross-sectional study to examine tolerogenic activity and associated gene expression in mDCs from 14 healthy controls (HCs) and 14 RR-MS patients (seven untreated and seven interferon beta (IFNβ)-treated). Patient samples were obtained from the outpatient MS clinic at the Brigham and Women’s Hospital (BWH). All the patients belong to the Comprehensive Longitudinal Investigations in MS at BWH Study (The CLIMB Study).

### 4.2. Standard Protocol Approvals, Registrations, and Patient Consents

Approval was granted by the local Ethical Committee under protocol # 1999P010435/BWH, and informed written consent was obtained from all patients.

### 4.3. mDC and Naïve T Cell Isolation

PBMCs were isolated from 100 mL of heparinized peripheral venous blood using Ficoll density gradient centrifugation. Total PBMCs were equally divided for DC and for autologous T cell isolation. The total myeloid DC population was isolated and negatively selected using a human Pan-DC enrichment kit (Miltenyi Biotec B. V & Co. KG, Bergisch Gladbach, Germany) according to the manufacturer’s protocol. Subsequently, the negatively selected DCs were stained with fluorochrome-conjugated antibodies for pDCs (Lin1^−^, CD11c^−^, CD123^+^) and mDCs (Lin1^−^, CD11c^+^, CD123^−^) (APC-CD11c-mAb, PE-CD123mAb, FITC-LIN1mAb-BD Biosciences) followed by FACS sorting (FACS ARIA- BD Bioscience, USA). Naïve CD4^+^ T cells were isolated via negative selection using a human naïve CD4^+^ T cell isolation kit (Miltenyi Biotec B. V & Co. KG, Germany) according to the manufacturer’s protocol and stored at 4 °C during the 24 h DC pretreatment until co-culture initiation.

### 4.4. In Vitro DC Stimulation and Culture

A portion of the FACS-sorted cells were used to isolate ex vivo RNA, while the remainder of the purified mDC were plated in 96-well plates (20,000 cells/well) in 200 μL of hematopoietic media X-VIVO15 (Lonza AG, Monteggio, Switzerland.) for 24 h with and without different combinations of stimuli: 100 ng/mL lipopolysaccharide (ecLPS 0111:B4; cat# L3012, Sigma–Aldrich, Saint Louis, MO, USA), 40 ng/mL recombinant human IL-27 (cat# 7229-10, Biovision, Hackensack, NJ, USA), 60 ng/mL rhIL-10 (cat# 57002, BioLegend Inc., San Diego, CA, USA), or no additional stimulation. After 24 h, the DCs were washed twice and then either lysed in RNA extraction buffer (RNA Purification Micro Kit, cat# 35300, Norgen Biotek Corp, Thorold, ON, Canada) or were co-cultured with autologous naïve CD4^+^ T-cells at a 5:1 ratio per well (T cells to mDCs) in 200 μL of X-VIVO15 media (Lonza AG, Visp, Switzerland) with 1 μg/mL of anti-human CD3mAb (Clone OKT3, NA/LE, BD Bioscience, San Jose, CA, USA). After 4 days, supernatants were collected and frozen (−80 °C) for future cytokine analysis and the cultures were pulsed with [^3^H]-thymidine for 18 h. For blocking experiments, we used mouse anti-human IL-10 antibody (cat#MAB2171, R&D system, San Diego, CA, USA) or the appropriate isotype control.

### 4.5. RNA Analysis

After extraction (RNA Purification Micro Kit, cat# 35300, Norgen Biotek Corp., Canada), the total RNA was analyzed in two ways. First, the RNA was analyzed directly using the nanostring hybridization platform (Human Immunology v2-Nanostring Technologies Inc, USA), containing 594 target genes, following manufacturer’s protocol. The data were analyzed using nSolver v.2.6 software (Nanostring Technologies, Inc., Washington, DC, USA). Background, calculated via means of all negative controls plus two standard deviations, was subtracted during analysis. Gene expression was normalized using the geometric mean of the positive controls and housekeeping genes in the assay. Quality control was based on normalization factors as per manufacture’s protocol, and no sample failed the quality control. Second, the RNA was reverse transcribed into cDNA and gene expression was determined using RT-qPCR for selected genes. The validated PCR primer/probe TaqMan PCR-specific human primer sequences were used for [*PDCD1LG2 (PD-L2), CD274 (PD-L1), IDO-1, SOCS1, IRF1, IL6R, STAT1, STAT3, STAT5A, IL27Rα, IL10, CSF2BR* (GM-CSF receptor)] (Applied Biosystem, Longmont, CO, USA), using *GADPH* as the housekeeping gene used to normalize the gene expression data.

### 4.6. Flow Cytometry Analysis

In some assays, the mDCs were harvested after 1 day of in vitro culture, then fixed and permeabilized with Cytofix/Cytoperm (BD Bioscience) following manufacturer protocol. The cells were stained with antibodies against human IDO1-PE (Ebioscience) and CD273-BV421 (anti-PD-L2, BD Bioscience) or isotype control Abs in BD Horizon Brilliant Stain Buffer following manufacturer protocol. The samples were acquired on an MACSQuant VYB flow cytometer (Miltenyl Biotec B. V & Co. KG, Germany) and analyzed using FlowJo v.10 software (FlowJo, LLC, Ashland, ON, USA).

### 4.7. Measuring Supernatant Cytokines

Supernants from the co-culture assays (mDCs with Naïve T-cells), were analyzed using the Milliplex Map Kit (Luminex, EMD Millipore Corporation, Burlington, MA, USA) for the following cytokines [Human Th17 Magnetic Bead Panel, cat# HTH17MAG-14K: IL-17F, IL-17A, IL-22, IL-21, IL-23, IL-27; Human CD8+ T Cell Magnetic Bead Panel, cat# HCD8MAG-15K: sCD137, Granzyme B, sFAS, sFASL, Perforin; and Human Cytokine/Chemokine Panel, cat# MPXHCYTO-60K: IFN-alpha2, IFN-gamma, IL-1beta, IL-6, IL-12 (p70), TNF alpha]. Samples were thawed on ice and processed according to the manufacter protocol. Plates were run on Luminex 200 with XPONENT software. We set the values for the undetected samples of each cytokine as the half of the minimal limit of detection by the standard curve.

### 4.8. Statistical Analysis

The studied variables were displayed according to frequency distribution. Data were analyzed using Prism 6 (GraphPad Software, LLC, San Diego, CA, USA). The statistical significance of differences was determined by two sample *t*-tests or ANOVAs depending on the assay; when an assumption of normality was not reasonable, a Wilcoxon test was performed. For all analyses, a two-tail *p*-value less than 0.05 was considered statistically significant.

## Figures and Tables

**Figure 1 ijms-24-08000-f001:**
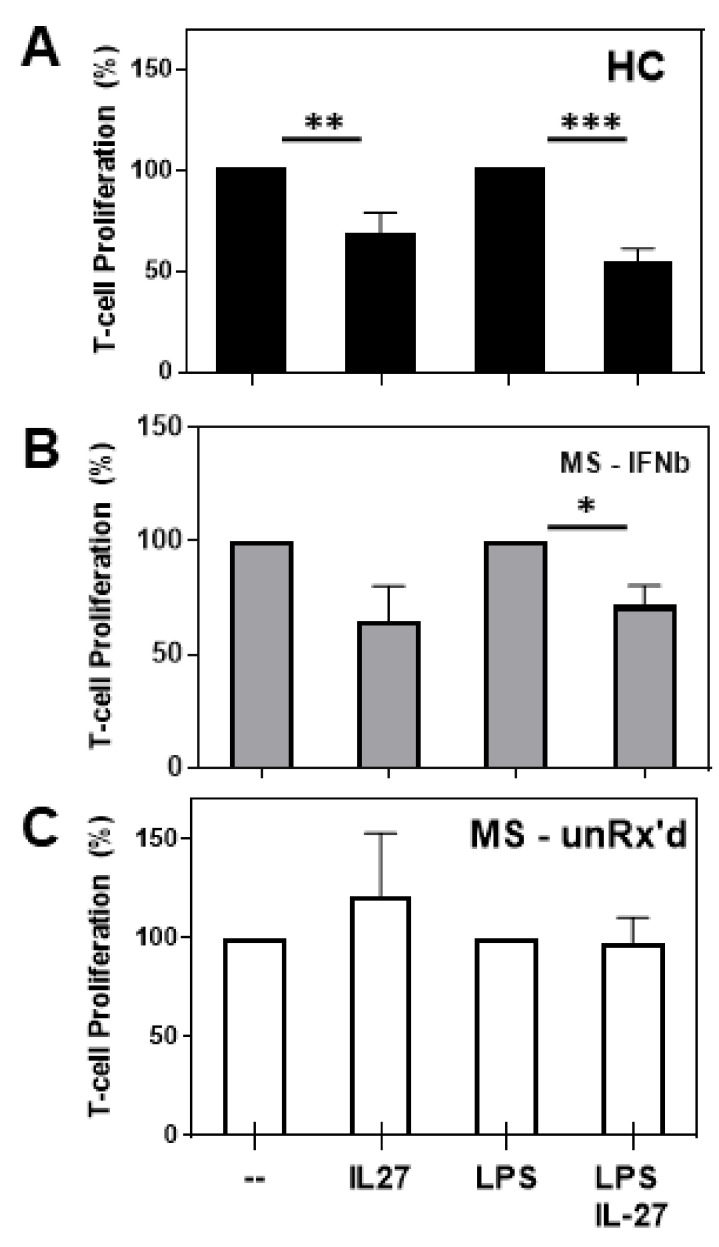
mDCs from HCs and IFNb-treated patients respond to IL-27 with tolerogenic activity. Freshly isolated mDCs, isolated from (**A**) HCs, (**B**) IFNb-treated MS patients, and (**C**) untreated patients with MS, were pretreated with IL-27, LPS, or LPS/IL-27 for 24 h before being co-cultured with autologous naïve CD4 T cells (at 1:5 mDC:T cell ratio) with anti-CD3 stimulation. Proliferation was measured after 4 days by overnight pulse with H^3^-thymidine. The proliferation of the co-cultures containing the mDCs treated with media or LPS were set to 100 percent for comparison to mDCs treated with IL-27 or IL-27/LPS, respectively. Significant changes in T cell proliferation were determined. (* *p* < 0.05, ** *p* < 0.01, *** *p* < 0.001).

**Figure 2 ijms-24-08000-f002:**
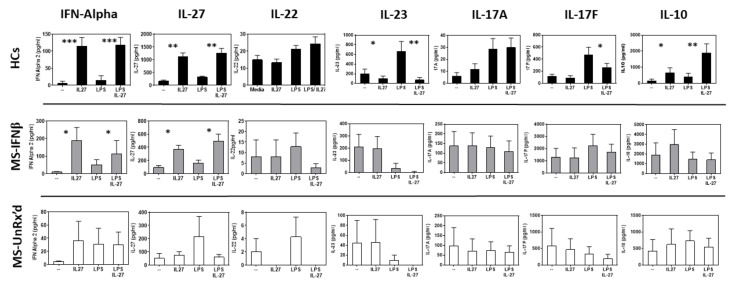
The ability of IL-27-treated mDCs to mediate T cell inhibition is associated with similar patterns of the secretion of innate cell-derived IFN-a and IL-27, while the majority of the T cell-derived cytokines segregate with disease status. Supernatants from the co-cultures shown in Figure 1 were analyzed for secreted cytokines using Luminex analysis. The results are organized to demonstrate the similarity of the HC and IFNb-treated cultures on the left and the similarities between the untreated and IFNb-treated MS samples to the right. Significant changes in cytokine secretion were determined using an analysis of variance (ANOVA) test (* *p* < 0.05, ** *p* < 0.01, *** *p* < 0.001).

**Figure 3 ijms-24-08000-f003:**
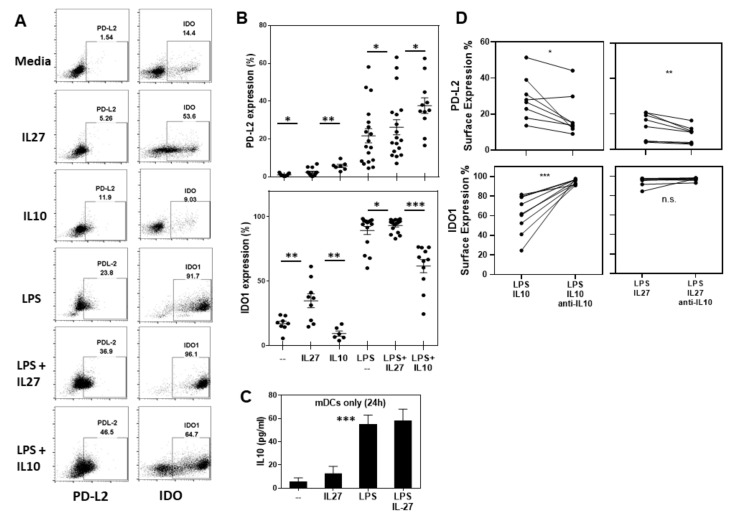
IL27 induces the expression of PD-L2 and IDO in HC mDCs: (**A**) fluorescence-activated single-cell sorting (FACS) plots show the representative expression of PD-L2 (surface) and IDO (intracellular) on HC mDCs in response to 24 h incubation with different stimuli. (**B**) Reproducibility of the capacity of IL-27 and IL-10 to induce PD-L2 and IDO is shown for multiple donors (n = 17). (**C**) The amount of IL-10 secreted after 24 h mDC stimulation with IL-27, LPS, or IL-27/LPS is shown, demonstrating significant changes in the mDC expression of IDO1 and PD-L2. (**D**) Neutralizing IL-10 in mDC cultures supplemented with LPS+IL-27 as well as LPS+IL10 resulted in a significant reduction in the level of PD-L2 expressed by the mDCs. IDO1 expression increased after neutralizing IL10 in the LPS+IL10 condition but did not change in the LPS+IL27 cell culture (n.s. not significant, * *p* < 0.05, ** *p* < 0.01, *** *p* < 0.001).

**Figure 4 ijms-24-08000-f004:**
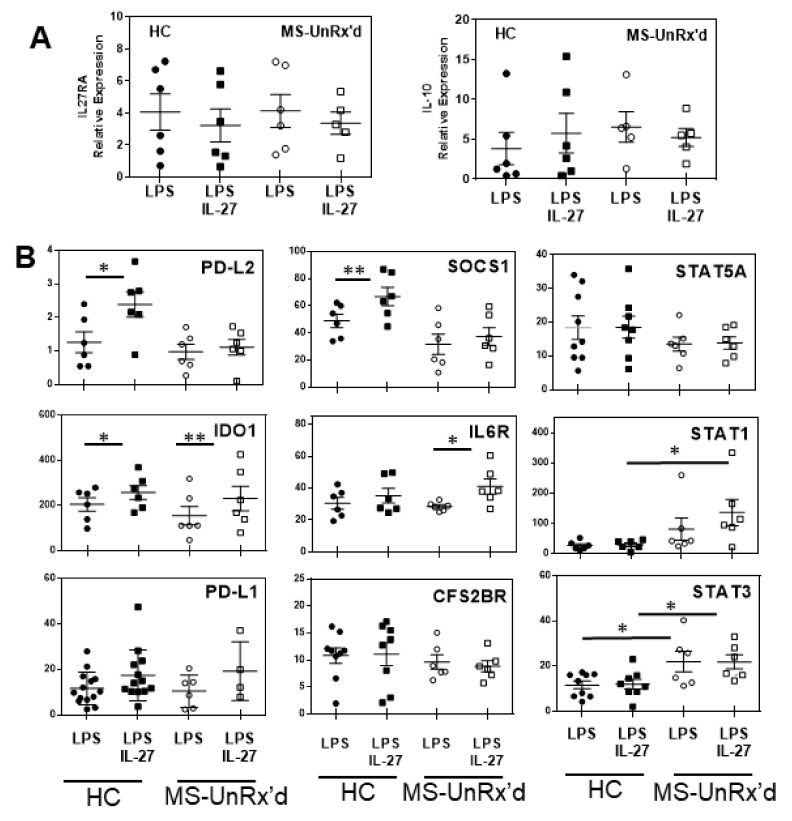
IL-27 differentially induces the expression of a number of pro- and anti-inflammatory genes in mDCs from healthy donors versus those from untreated patients with MS. How IL-27 treatment affects mDC expression of a panel of genes was determined using Taqman qPCR analysis of RNA isolated from HC- (n = 6) or untreated MS patient- (n = 6) derived mDCs that had been stimulated (24 h) with LPS or LPS/IL-27. (**A**) Two genes that are well-known to regulate IL-27 responses, IL-27Ra and IL-10, are shown. (**B**) The capacity of IL-27 to affect the expression of PD-L2, IDO1, PD-L1, SOCS1, IL6R, CFS2BR, STAT5A, STAT1, and STAT3 is shown (data are relative to GADPH). Filled symbols = HCs; Open symbols = untreated MS patients (* *p* < 0.05, ** *p* < 0.01).

**Figure 5 ijms-24-08000-f005:**
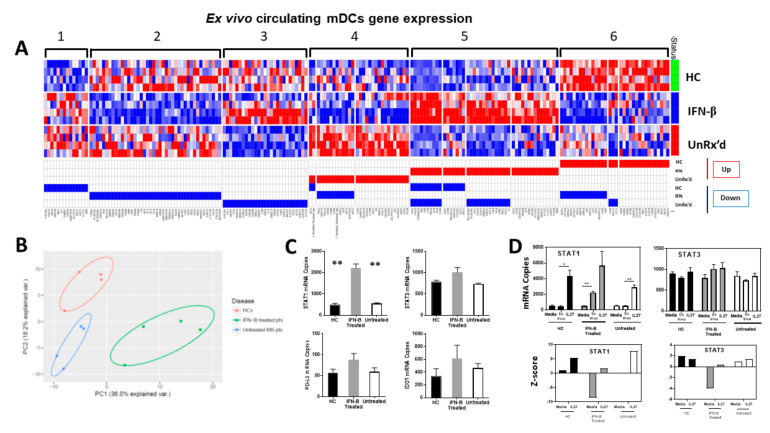
Direct analysis of mRNA transcript frequency containing 594 target genes (Human Immunology v2-Nanostring Technologies) in HC, IFNb-treated MS, and untreated MS-derived mDCs indicates six IL-27 response transcription patterns. Each mDC sample was tested using Hybridization gene expression analysis ex vivo (time 0) and after 24 h with versus without IL-27 stimulation: (**A**) Heatmap analysis showing the resulting six gene expression patterns (I through VI). mDCs gene expression in the natural state (time 0) was compared to the genes’ expression after treatment with versus without IL27 (Asymptotic *t*-test). (**B**) PCA plot analysis showing that mDC gene expression resulted in disease- and treatment-specific clustering. (**C**) Individual analysis of selected mRNA transcripts (STAT1, STAT3, PDL2, and IDO) from natural circulating mDCs. (**D**) How the HC, IFNb-treated MS, and untreated mDCs compared for STAT 1 and STAT 3 genes as indicated in the nanostring analysis. The data show the changes in transcript numbers with 24 h in media, ex vivo (time 0), and 24 h with IL-27, with the respective Z-score from the hybridization gene expression assay (* *p* < 0.05, ** *p* < 0.01).

## Data Availability

The dataset used and analyzed during the current study is available from the corresponding author on reasonable request.

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
