# Peer review of "Defective Induction of IL-27-Mediated Immunoregulation by Myeloid DCs in Multiple Sclerosis"

_ijms, 2023, doi:10.3390/ijms24098000_

Round 1
Reviewer 1 Report
Von Glehn et al. provide evidence that mDC isolated from MS patient treated with IFNb and from healthy donors present a tolerogenic phenotype in response to IL-27. They provide insights on cellular pathways associated to disease and treatment mechanisms. They integrate correctly their new results in the scientific corpus with their introduction and their conclusion.
Comments:
In the introduction, the authors should justify why they choose to work on mDC specifically instead of other DC subtypes. For example, cDC2 can also induce tolerogenicity but more potent APC than mDC.
mDC were adequately isolated and sorted from PMBC. However, the authors cultured them up to 4 days in some case, without cytokines such as GM-CSF and IL-4 to help survival and differentiation while doing functionnal assays (in-vitro T cell activation). The authors should provide cytometry evidence that the mDC survived after this period and still possess mDC markers.
The authors used H3 thymidine incorporation to measure T cell proliferation. This method as been slowly replaced over the last decade with more precise methods such as cell labeling with CSFE, CTV or equivalent molecules. Doing that, the authors would provide information of the actual number of proliferating T cells and number of generation instead of a proliferation index based on a marker relative abundance. Currently, the reader as no idea if mDC could induce significant T cell proliferation based on normalized data of thymidine detection.
The authors have provided evidence of the expression of co-inhibitory molecules in mDC. Would it be relevant to also show the expression of APC co-stimulatory molecules (CD80, CD86, OX-40L, for example).
In the current version of Figure 5A, the gene names at the bottom of the heatmap are unreadable due to the low resolution. Even if the resolution can be improved, authors should provide the list as a supplemental table.
Author Response
Reviewer #1 comments
- In the introduction, the authors should justify why they choose to work on mDC specifically instead of other DC subtypes. For example, cDC2 can also induce tolerogenicity but more potent APC than mDC.
Response: As the frequency of natural DCs is around 1% of peripheral blood mononuclear cells (PBMCs), there are not many publications that have studied activities exhibited by circulating human dendritic cells. Thus, most publications test the activities of monocyte-derived DCs that were induced to become Mo-DCs by culturing them with various cytokine mixtures (Von Bubnoff D, 2011). In this report, we had wanted to bypass the question as to whether the activities of induced Mo-DCs actually reflect those of ex vivo isolated DCs, especially since we wanted to test whether DC function would change after a 24-hour incubation with the cytokine IL-27. As a result, while we would have preferred to subdivide total ex vivo mDC’s into the mDC1 [CD1c+ BDCA1] and mDC2 [CD141+ BDCA3] constituents, the low number of mDCs acquired after sorting necessitated that we use the whole mDCs population. More specifically, our post-FACs sorting total mDC yield (as FACS sorting optimally gives less than 40% of the theretical yield) from the ~1.5x10-8 PBMCs/sample, representing entire 10 tubes of freshly donated blood, were only enough to perform triplicate T cell co-culture assays and to provide RNA samples for analyzing ex vivo (time 0) and 24 hour response to the different stimuli (+/- LPS, +/- IL-27), as reproducible nanostring data requires input from 20,000 cells. Thus, for this initial research, it was not possible to further dissect the mDC population. However, now that we have identified this novel tolerogenic activity that occurs in response to IL-27 treatment that is differentially induced depending upon patient treatment (MS,-IFNb v MS-UnRx’d), our plan is to focus on the mDC1/2 subtypes in the future (and use single cell analyses that will allow reduced cell numbers).
- mDC were adequately isolated and sorted from PMBC. However, the authors cultured them up to 4 days in some case, without cytokines such as GM-CSF and IL-4 to help survival and differentiation while doing functional assays (in-vitro T cell activation). The authors should provide cytometry evidence that the mDC survived after this period and still possess mDC markers.
Response: We agree that changes in viability of isolated and cultured cDCs have been under-appreciated in the published literature and could be a confounding variable in long-term in vitro studies. In their capacity to induce T cell hypo-responsiveness, tolerogenic DCs have been proposed either to be unable to provide co-stimulatory signals or to actively provide dominant negative signals (Gallucci et all 1999, Nat Med 5: 1249–1255). Fundamentally, the issue raised in this reviewer comment is whether the mDCs are equally viable and functional after the 24hr pretreatment, or whether the IL-27 treated DCs are non-viable and thus are non-stimulatory rather than tolerogenic. As even tolerizing DCs are known to initially allow a certain amount of T cell proliferation (Steinman et al. Annals of NY Academy of Sciences 2003; 987:15-25), one way to measure DC viability would be to ascertain the capacity of each DC population to induce the initial activation of naïve T cells, a process requiring immunological synapse formation and TCR crosslinking (Lee et al 2002, Science 295, 1539-1542; Tseng et al 2008, J. Immunol. 181, 4852-4863.). Thus, we theorized that if the cDC1/cDC2 underwent significant apoptosis in our 24hour pre-treatment cultures, they would exhibit a severely reduced capacity to mediate initial T cell activation which is associated with their release of sCD137, sFASL, and extracellular Granzyme B. The data in supplemental figure 2 show that the differently treated HC, MS-IFNb, and MS-UnRx’d mDCs exhibit equal capacity to induce co-stimulation and synapse formation as indicated by the similar and high detection of T cell derived sCD137 sFASL, and GzmB. Thus, we believe that these data indicate that all of the 24hr pretreated DCs are able to similarly engage and activate the T cells to a certain extent, whether they exhibit altered viability by the end of the four-day culture is a great question that we will include in our future studies and is a possibility that we have added to the text. Regardless, if the UnRx’d MS mDCs lack tolerogenic activity because they preferentially undergo apopotosis in response to IL-27– that in itself would be a striking mechanism.
- The authors used H3 thymidine incorporation to measure T cell proliferation. This method as been slowly replaced over the last decade with more precise methods such as cell labeling with CSFE, CTV or equivalent molecules. Doing that, the authors would provide information of the actual number of proliferating T cells and number of generations instead of a proliferation index based on a marker relative abundance. Currently, the reader as no idea if mDC could induce significant T cell proliferation based on normalized data of thymidine detection.
Response: We agree with reviewer’s comments that the data presentation as % of T cell proliferation does not allow the reader to understand the magnitude of the proliferation that occurred under each condition. Thus, as per the suggestion, we have added Supplemental Figure 1 which shows the extent of proliferation in the cultures. While these data still utilize H3 thymidine incorporation, these raw data indicate that the culture conditions induced strong proliferation in the control cultures, where the addition of IL-27 to the corresponding cultures, did (HC/MS-IFNb) or did not (MS-unRx’d) significantly reduce proliferation.
- The authors have provided evidence of the expression of co-inhibitory molecules in mDC. Would it be relevant to also show the expression of APC co-stimulatory molecules (CD80, CD86, OX-40L, for example).
Response: We agree with reviewer’s suggestion and added a supplemental figure (supplemental figure 5) showing how the HC, IFNb-treated MS, and Untreated mDCs compared for the expression of APC co-stimulatory molecules (CD80, CD83, CD86, CCL2 and CD40 genes) as indicated in the nanostring analysis. The data show the changes in transcript numbers with 24 hrs in media, ex vivo (time 0), and 24 hrs with IL-27, with the respective Z-score from the hybridization gene expression. In HC, we observed an increased expression of CD80 mRNA induced by IL27 treatment. It was not observed in MS patients. Untreated MS patients had a significant redution of CD86 mRNA expression after IL27 treatment.
In the current version of Figure 5A, the gene names at the bottom of the heatmap are unreadable due to the low resolution. Even if the resolution can be improved, authors should provide the list as a supplemental table.
Response: The list will be inserted at supplemental material.
Reviewer 2 Report
Recommendation: Major revisions.
In this manuscript, Felipe von Glehn. Et al tested the effect of IL27 on myeloid dendritic cells (mDCs) from patients with MS and healthy donors (HC). This work aimed to figure out whether IL27 could produce different tolerogenic responses on mDCs derived from IFNb-treated/untreated patients and in healthy donors. The authors based their hypothesis on a few previous works that have described the modulator properties of IL27 on mDCS.
The authors have done an interesting descriptive and mechanist work, which shows compelling findings in vitro related to the behavior of mDCs derived from MS patients. In brief, the authors show that mDCs cultured with naïve cells from healthy patients treated with IL27 inhibited T cell proliferation, increased anti-inflammatory cytokine released such as IL27, and IL10, and augmented the expression of tolerogenic proteins like PD-L2, IDO1, and SOCS1 compared with mDCs from MS-untreated patients, which were unresponsiveness upon IL27. Strikingly, the authors observed that mDCs from IFNb-treated-patients performed the same protolerogenic effect upon IL27 as healthy patients, suggesting that IFNb may regulate IL-27 responsiveness, playing a role in the induction of tolerogenic mDC function.
The authors have done a great work and their findings may be of importance in the field. However, there are some important questions related to the stimulation of mDCS with LPS that the authors should be addressed to clarify and improve these outcomes before being published.
Comments and suggestions:
The most important concern is the use of LPS on mDCS stimulation. Although LPS is a well-described macromolecule derived from bacteria that activates APCs, it is not a good option in the context of Multiple Sclerosis. One of the things that draw attention is the unresponsiveness effect on T cell proliferation (fig 1) after LPS stimulation. LPS has been described to promote effector t cells (th1) through mDCS activation (For example J Exp Med (2000) 192 (3): 405–412 or https://doi.org/10.1073/pnas.0500098102). The authors should have observed changes in T-cell proliferation in these cultures. However, the authors do not see differences (LPS vs no LPS (control, fig 1). Furthermore, it is known LPS acts directly in T cells, producing their activation (J Exp Med. 1997 Jun 16; 185(12): 2089–2094, Crit Rev Immunol. Author manuscript; available in PMC 2013 Jan 21, https://doi.org/10.1371/journal.pone.0144375). The design of these experiments doesn´t allow for distinguishing whether the effect on T cell proliferation proceeds from mDCS or T cells themselves. Even though the effect of IL27 on mDCs without LPS (fig 1) would be enough to explain the tolerogenic effects on these experiments. The most important concern is regarding the release or expression of IL27, IL10, SOCS, and IDO1 after LPS stimulation (figs 2 and 3). Due to it being of great importance experiments to support the mechanistic hypothesis of IL27 on mDCS, the authors should perform additional experiments using a specific MS antigen. A suggestion for the authors is to use the MOG35-55 peptide (Ann Neurol. 2009 Apr; 65(4): 457–469.) or other myelin-derived peptides usually used in cultures (APCs/naïve T cells) in MS experiments.
The authors should perform and add to all of figures 1, 2, and 3 the data they will obtain upon stimulation of mDCS with this peptide, showing the effect on T cell proliferation, cytokine release, and the synergistic effect with IL27 and Il10 on IDO and PDL2 expression.
Author Response
Reviewer #2 comments
1.The most important concern is the use of LPS on mDCS stimulation. Although LPS is a well-described macromolecule derived from bacteria that activates APCs, it is not a good option in the context of Multiple Sclerosis. One of the things that draw attention is the unresponsiveness effect on T cell proliferation (fig 1) after LPS stimulation. LPS has been described to promote effector t cells (th1) through mDCS activation (For example J Exp Med (2000) 192 (3): 405–412 or https://doi.org/10.1073/pnas.0500098102). The authors should have observed changes in T-cell proliferation in these cultures. However, the authors do not see differences (LPS vs no LPS (control, fig 1). Furthermore, it is known LPS acts directly in T cells, producing their activation (J Exp Med. 1997 Jun 16; 185(12): 2089–2094, Crit Rev Immunol. Author manuscript; available in PMC 2013 Jan 21, https://doi.org/10.1371/journal.pone.0144375). The design of these experiments doesn´t allow for distinguishing whether the effect on T cell proliferation proceeds from mDCS or T cells themselves. Even though the effect of IL27 on mDCs without LPS (fig 1) would be enough to explain the tolerogenic effects on these experiments. The most important concern is regarding the release or expression of IL27, IL10, SOCS, and IDO1 after LPS stimulation (figs 2 and 3). Due to it being of great importance experiments to support the mechanistic hypothesis of IL27 on mDCS, the authors should perform additional experiments using a specific MS antigen. A suggestion for the authors is to use the MOG35-55 peptide (Ann Neurol. 2009 Apr; 65(4): 457–469.) or other myelin-derived peptides usually used in cultures (APCs/naïve T cells) in MS experiments.
The authors should perform and add to all of figures 1, 2, and 3 the data they will obtain upon stimulation of mDCS with this peptide, showing the effect on T cell proliferation, cytokine release, and the synergistic effect with IL27 and Il10 on IDO and PDL2 expression.
Response: Thank you for your suggestion. The figure 1 demonstrated the effect of IL27 compared to media (media vs. IL27) and compared to LPS (LPS vs LPS+IL27). Thus, the proliferation was set to 100 percent for comparison to mDCs treated with IL-27 or IL-27/LPS respectively. Thus, the actual proliferation values were not presented. To clarify this point, we provided a supplemental figure showing H3 thymidine incorporation of each corresponding condition demonstrated at figure 1. Thus, you can see that there was marked proliferation by the T cells in all conditions – although it was significantly lower in the cultures containing HC or MS-IFNb mDCs that had been pre-treated with IL-27.
Regarding the request to use a defined antigen such as MOG 35-55 that has shown widespread use and given much information in mouse EAE models when used as an immunogen with CFA: unfortunately, the use such a defined single peptide antigen in ex vivo human studies is problematic. MS is a very complex disease with undefined etiology. Multiple CNS-derived proteins thought to may play a role in different individuals potentially in conjunction with their individual HLA alleles. Moreover, the proposed peptide loading of the mDCs for specific antigen presentation will suffer from the low frequency of antigen specific T cells in the blood. Previous research has demonstrated that MBP specific T cells are not only detected in both healthy donors and MS patients with MS, but they are present at an extremely low frequency of roughly 1/1x106 T cells. Thus, it would be impossible to generate enough mDC-T cell cultures to cover that low frequency event. However, we did try to use a stimulus that would require direct interaction between the DC and the T cells by providing the anti-CD3 antibody in a soluble rather than a plate- or bead-bound format and did not give anti-CD28 co-stimulatory antibody. Therefore, it is unfortunate, but the strategy of specific mDCs stimulation it’s not feasible due to (i) low number of mDCs recovered by the peripheral blood; (ii) difficulty to find untreated MS patients; (iii) difficulty to find a universal myelin derived peptide to be used in all MS patients.
Round 2
Reviewer 1 Report
The authors have adequately answered my concerns and corrected the small mistakes found during the 1rst round of revision.
In the legend of supplemental figure 1, authors refers to part A, B and C of the figure, letters that don't appears on the actual figure.
The manuscript should be accepted in this state after this small modification
Reviewer 2 Report
Thank you to the authors for the prompt response.
The authors have clarified all of my concerns.
I recommend this article to publish in IJMS